# MATH-Perturb: Benchmarking LLMs' Math Reasoning Abilities against Hard Perturbations

**Kaixuan Huang** [1]  **Jiacheng Guo** [†1]  **Zihao Li** [†1]  **Xiang Ji** [†1]  **Jiawei Ge** [†1]  **Wenzhe Li** [†1]  **Yingqing Guo** [†1]
**Tianle Cai** [†1]  **Hui Yuan** [†1]  **Runzhe Wang** [†1]  **Yue Wu** [†1]  **Ming Yin** [†1]  **Shange Tang** [†1]
**Yangsibo Huang** [2]  **Chi Jin** [1]  **Xinyun Chen** [2]  **Chiyuan Zhang** [2]  **Mengdi Wang** [1]

## Abstract

Large language models have demonstrated impressive performance on challenging mathematical reasoning tasks, which has triggered the discussion of whether the performance is achieved by true reasoning capability or memorization. To investigate this question, prior work has constructed mathematical benchmarks when questions undergo *simple perturbations* – modifications that still preserve the underlying reasoning patterns of the solutions. However, no work has explored *hard perturbations*, which fundamentally change the nature of the problem so that the original solution steps do not apply. To bridge the gap, we construct MATH-P-Simple and MATH-P-Hard via simple perturbation and hard perturbation, respectively. Each consists of 279 perturbed math problems derived from level-5 (hardest) problems in the MATH dataset (Hendrycks et al., 2021). We observe significant performance drops on MATH-P-Hard across various models, including o1-mini ($-16.49\%$) and gemini-2.0-flash-thinking ($-12.9\%$). We also raise concerns about a novel form of memorization where models blindly apply learned problem-solving skills without assessing their applicability to modified contexts. This issue is amplified when using original problems for in-context learning. We call for research efforts to address this challenge, which is critical for developing more robust and reliable reasoning models. The project is available here.

## 1. Introduction

Large language models (LLMs) have achieved remarkable progress in solving many previously challenging tasks and demonstrating signs of general intelligence (Bubeck et al., 2023). As LLMs become more intelligent, the research community responds by developing and adopting new benchmarks to guide the development of better models (Wang et al., 2024; Zhou et al., 2023; Liu et al., 2024; Rein et al., 2023; Yan et al., 2024).

In mathematical reasoning, the field has progressed from simpler datasets like SVAMP (Patel et al., 2021) and GSM8K (Cobbe et al., 2021a) to more challenging benchmarks such as MATH (Hendrycks et al., 2021), Olympiad-Bench (He et al., 2024), and AIME problems. Models continue to strike higher performance on these advanced benchmarks through stronger architectures, novel training approaches, and better training data (OpenAI, 2024; Yang et al., 2024; Shao et al., 2024; DeepSeek-AI et al., 2025).

Nevertheless, concerns about data contamination and out-of-distribution generalization remain. Model performance can be artificially high if variants of the evaluation set leak into the training datasets or if its distribution is over-represented. In these cases, the model could be merely doing pattern recognition and memorizing the solution steps without understanding the underlying rationale, making it vulnerable to perturbations of the problem formulation (Zhang et al., 2024; Srivastava et al., 2024).

Several works have been proposed to quantify the robustness of reasoning models against such perturbations (Shi et al., 2023a; Mirzadeh et al., 2024; Zhang et al., 2024; Srivastava et al., 2024; Gulati et al., 2024; Zou et al., 2024). Notably, Srivastava et al. (2024) created Functional-MATH by manually rewriting the original problems in the MATH benchmark (Hendrycks et al., 2021) into problem templates, where the numerical values in the problem statements and the corresponding answers can be varied automatically to generate infinitely-many versions that use the same math problem-solving skills. They observed performance drops between the modified benchmark and the original benchmark for several state-of-the-art language models, indicating that those models are indeed *biased* towards the original configurations of numerical values due to some form of data contamination. However, most existing work focuses

---

[†]Core contribution  [1]Princeton University [2]Google. Correspondence to: Kaixuan Huang <kaixuanh@princeton.edu>.

*Proceedings of the $42^{nd}$ International Conference on Machine Learning*, Vancouver, Canada. PMLR 267, 2025. Copyright 2025 by the author(s).

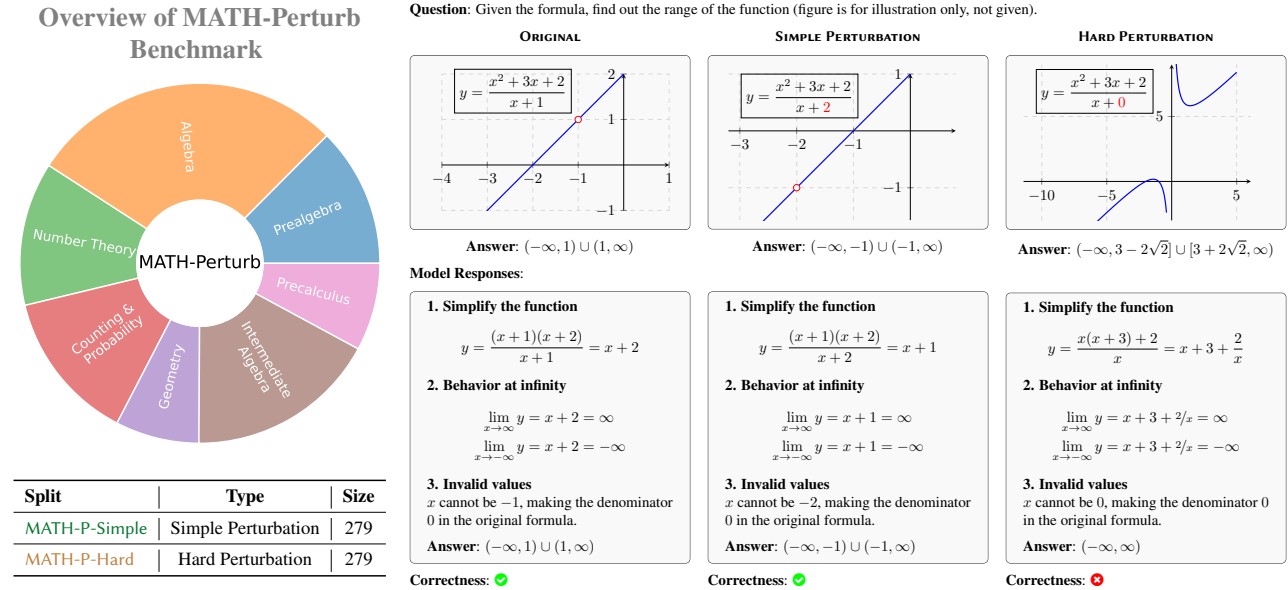

*Figure 1.* **Left:** The overview of MATH-Perturb Benchmark. **Right:** An example of the original problem, its **simple** perturbation, its **hard** perturbation, and the corresponding model responses that overfit the short-cut solution. The simple perturbation to the problem is non-essential, so the modified problem can be solved using the same method as the original problem. The hard perturbation changes the problem fundamentally and it requires more difficult problem-solving skills. The shortcut solution can solve the original problem and its simple perturbation but fails on the hard perturbation.

on perturbing non-critical parameters (e.g., numerical values) that do not alter the fundamental reasoning patterns required to solve the problem. We refer to such changes as **simple perturbations**. While prior studies have shown that LLMs can generalize across a range of problem variants by relying on bag-of-heuristics reasoning (Nikankin et al., 2024; jylin04 et al.), this form of generalization does not necessarily reflect a true understanding of the underlying principles. As a result, models may still fail when faced with a substantial shift in reasoning patterns.

In this work, we take one step forward beyond simple perturbations. We consider **hard perturbations**: while at lexical level (e.g. edit distance) the modification is similar to simple perturbations, we ensure to change the problem formulations fundamentally so that the original solution paths are no longer applicable to the perturbed settings; see Figure 1 for a comparison between the two types of perturbations. A genuinely robust reasoning model that understands the underlying rationales should *not only* solve the modified problems under simple perturbations *but also* be able to judge whether the problem formulations change in a way that fundamentally alters the problems and respond accordingly, instead of applying the learned skills indiscriminately.

As the capabilities of large language models continue to advance and the average-case performance continue to improve, the generalization abilities against hard perturbations may soon become the primary bottleneck in their real-world usages. Addressing this challenge will be critical for advancing the robustness and reliability of future LLMs.

We summarize our contributions and key findings below:

- We design and construct MATH-P-Simple (simple perturbation) and MATH-P-Hard (hard perturbation), each consisting of 279 perturbed math problems that originate from the level-5 (hardest) problems of the MATH dataset (Hendrycks et al., 2021). The datasets are curated by 12 graduate-level experts with rigorous rubrics and cross-checking workflow for quality control (Section 2).

- We benchmark the math reasoning abilities of 18 LLMs (Section 3.1), and show that all the models, including o1-mini and gemini-2.0-flash-thinking, suffer significant performance drops (10%-25%) on MATH-P-Hard. This indicates these models are biased towards the original distribution of reasoning patterns and suffer from *out-of-distribution effect* when facing problems with hard perturbations.

- We conduct in-depth failure mode analysis (Section 3.2) and identify a new form of memorization, where the model memorizes the problem-solving techniques from the training set and blindly applies them without judging whether the modified settings are still suitable.

- We investigate the influences of in-context learning (ICL) with the corresponding original unmodified problem and solution (Section 3.4), and demonstrate that ICL with original example may hurt the model on MATH-P-Hard, as the models may fail to recognize the subtle differences and get misled by the demonstration.

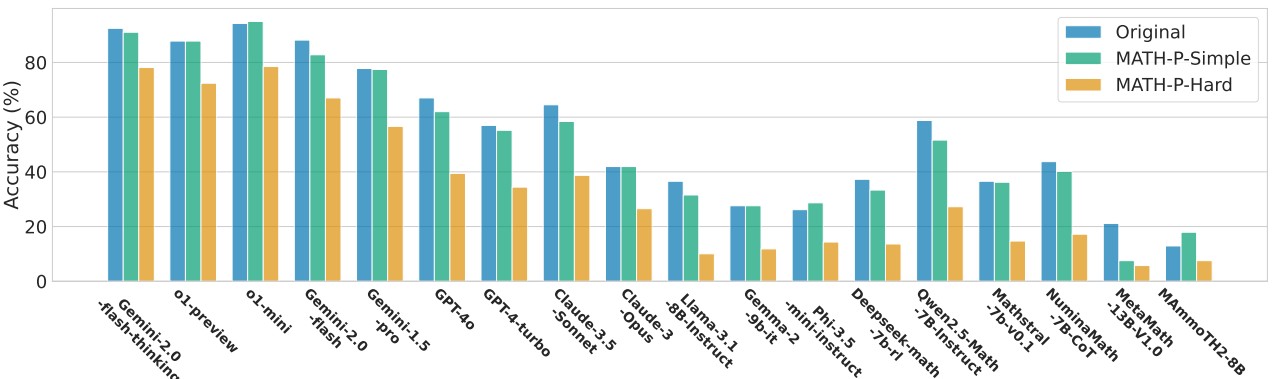

*Figure 2.* Performance on MATH-P-Simple, MATH-P-Hard, and the corresponding Original problems. We observe performance degradations across all models on MATH-P-Hard.

## 2. Dataset Curation

**Origin of the Dataset.** We choose the popular MATH benchmark (Hendrycks et al., 2021), which contains challenging mathematical reasoning problems sourced from American high school mathematics competitions such as the AMC 10, AMC 12, and AIME. Each problem belongs to one of the 7 subjects: Prealgebra, Algebra, Number Theory, Counting and Probability, Geometry, Intermediate Algebra, and Precalculus. Besides, each problem is labeled with a difficulty level of 1 (easiest) to 5 (hardest). The problems may contain LaTeX and Asymptote graphics language for describing mathematical concepts and geometric figures.

As the state-of-the-art reasoning models can already solve MATH problems with overall accuracies higher than 90% (OpenAI, 2024; Team et al., 2024a; DeepSeek-AI et al., 2025), we opt to focus only on the hardest level-5 problems in our work, and create new benchmarks from these level-5 problems. We use level-5 problems from both the `train` split and the `test` split as the seed problems, so we are able to investigate whether language models behave differently on the two splits.

**Annotation Criterion.** For each problem, we modify the problem to create two variations:

(1) for MATH-P-Simple, we make **simple perturbations**, i.e., non-essential modifications to the problem, ensuring that the modified problem can be solved using *the same method* as the original problem.

(2) for MATH-P-Hard, we make **hard perturbations**, i.e., small but fundamental modifications to the problem so that the modified problem *cannot* be solved using *the same method* as the original problem. Instead, it requires deeper math understanding and harder problem-solving skills.

Besides, we ensure the following two additional requirements:

- **Minimal Edits:** To test the generalization of the reason-

ing models and elicit potential memorization behaviors, we ask the annotators to make as minimal modifications as possible. Therefore, the modified problems stay close to the original problems in the text form.

- **Changed Answers:** For both of the modifications, we guarantee that the answers to the modified problems are different from the original answer. Therefore, models cannot cheat by pattern recognition and outputting memorized solutions.

**Quality Control.** We recruited 12 annotators (PhD students) with strong mathematical backgrounds for the annotation task. All the annotators hold a bachelor's degree in mathematics, have done researches in theoretical machine learning, and/or competed in mathematical competitions during high school.

To ensure the quality of the benchmark, all the annotators are required to double-check their annotations. Each modified problem is also cross-validated by an independent annotator to make sure the answer is correct.

Additionally, we manually went through all the problems where the o1-mini's answer and the annotated answer differ and confirmed that the annotated answers are correct.

**Benchmark Overview and Statistics.**

After removing several annotations that failed the quality checks, we obtained 279 pairs of modifications, where 164 examples are from `train` split and 115 examples are from `test` split. The numbers of problems in each of the 7 subjects are listed in Table 3. Figure 1 shows one example of our benchmark.

To quantify how similar the original problem and the modified problem are, first, we calculate the edit distance between the modified problem and the original problem, normalized by the length of the original problem. Besides, we compute the cosine similarities between the embeddings of the two problems, where we use OpenAI's

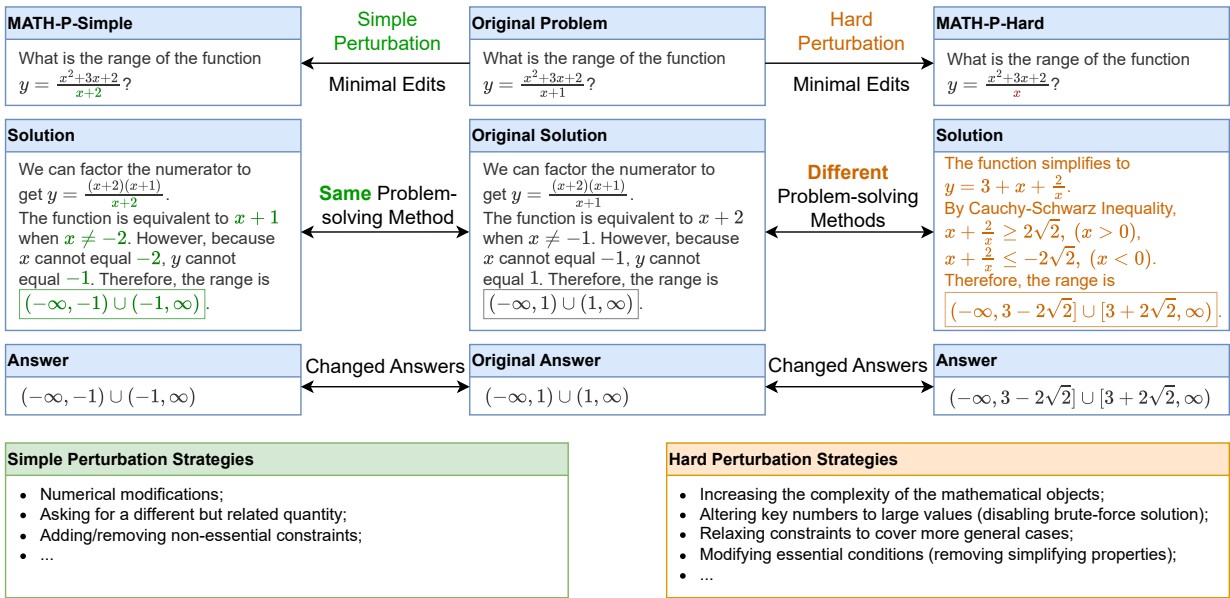

*Figure 3.* Illustration of the annotation process for MATH-P-Simple and MATH-P-Hard.

text-embedding-3-large embedding model. The distributions of the normalized edit distance and the cosine similarities are shown in Figure 4.

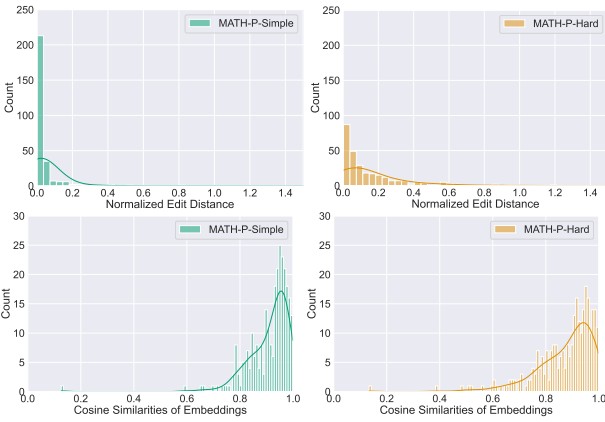

*Figure 4.* The distributions of edit distances and cosine similarities of embeddings between the perturbed problems and the original problems. The edit distances are normalized by the lengths of the original problems. The embedding model is OpenAI's text-embedding-3-large.

We also calculate the Mean Reciprocal Ranks (MRRs) when using the perturbed problem as the query to retrieve the corresponding original problem from the set of all 279 original problems, with the cosine similarities of embeddings being the ranking method. The MRRs of the MATH-P-Simple problems and MATH-P-Hard problems are 0.995 and 0.986, respectively, indicating that the corresponding original problem and solution are likely to be retrieved using typical semantic-based retrieval methods.

**Common Strategies for Perturbations.**

For MATH-P-Simple, most of the problems are modified numerically without making the problems fundamentally different. Our annotators have checked these numerical modifications are non-essential to the problems, so the modified problems can be solved with the same reasoning patterns. Besides, our annotators also adopt other types of changes. For example, asking for a different but related quantity, adding/removing non-essential constraints, and changing a mathematical concept to its contrasting counterpart.

For MATH-P-Hard, the modification strategies are more diverse and problem-specific. A general strategy is to increase the complexity of the mathematical objects involved. For example, raising the degrees of polynomials will make them harder to solve or factorize. Altering key numbers to large values can make brute-force solutions infeasible. Instead, solving the problem requires deriving general formulas or applying deeper theorems rather than relying on computational shortcuts. Other common strategies include relaxing constraints to cover more general cases, changing essential conditions so the original simplifying properties (e.g. symmetry, reducibility, linearity) no longer hold.

## 3. Experimental Results

**Evaluation Setting.** We adopt zero-shot chain-of-thought (CoT) (Wei et al., 2022; Kojima et al., 2022) as the standard evaluation method on our benchmarks. For comparison, we also evaluate the models on the set of the original 279 prob-

*Table 1.* Zero-shot CoT performance of the LLMs (accuracy, %). Original refers to the set of 279 unmodified problems. For the `train` and `test` columns, we report the accuracies for problems that *originate* from the `train` split and `test` split, respectively.

| Model | Original | | | MATH-P-Simple | | | MATH-P-Hard | | |
|---|---|---|---|---|---|---|---|---|---|
| | All | train | test | All | train | test | All | train | test |
| Gemini-2.0-flash-thinking-exp | 92.47 | 92.68 | 92.17 | 91.04 | 87.80 | 95.65 | 78.14 | 77.44 | 79.13 |
| o1-preview | 87.81 | 88.41 | 86.96 | 87.81 | 87.80 | 87.83 | 72.40 | 73.78 | 70.43 |
| o1-mini | 94.27 | 93.90 | 94.78 | 94.98 | 93.29 | 97.39 | 78.49 | 79.27 | 77.39 |
| Gemini-2.0-flash-exp | 88.17 | 87.20 | 89.57 | 82.80 | 81.71 | 84.35 | 67.03 | 68.29 | 65.22 |
| Gemini-1.5-pro | 77.78 | 77.44 | 78.26 | 77.42 | 76.83 | 78.26 | 56.63 | 56.10 | 57.39 |
| GPT-4o | 67.03 | 68.90 | 64.35 | 62.01 | 60.98 | 63.48 | 39.43 | 37.80 | 41.74 |
| GPT-4-turbo | 56.99 | 55.49 | 59.13 | 55.20 | 56.71 | 53.04 | 34.41 | 36.59 | 31.30 |
| Claude-3.5-Sonnet | 64.52 | 62.80 | 66.96 | 58.42 | 57.32 | 60.00 | 38.71 | 38.41 | 39.13 |
| Claude-3-Opus | 41.94 | 39.02 | 46.09 | 41.94 | 39.63 | 45.22 | 26.52 | 25.00 | 28.70 |
| Llama-3.1-8B-Instruct | 36.56 | 45.12 | 24.35 | 31.54 | 35.37 | 26.09 | 10.04 | 10.98 | 8.70 |
| Gemma-2-9b-it | 27.60 | 28.05 | 26.96 | 27.60 | 30.49 | 23.48 | 11.83 | 12.80 | 10.43 |
| Phi-3.5-mini-instruct | 26.16 | 27.44 | 24.35 | 28.67 | 26.83 | 31.30 | 14.34 | 15.24 | 13.04 |
| Deepseek-math-7b-rl | 37.28 | 42.68 | 29.57 | 33.33 | 35.37 | 30.43 | 13.62 | 15.85 | 10.43 |
| Qwen2.5-Math-7B-Instruct | 58.78 | 59.15 | 58.26 | 51.61 | 50.00 | 53.91 | 27.24 | 29.88 | 23.48 |
| Mathstral-7b-v0.1 | 36.56 | 43.29 | 26.96 | 36.20 | 42.07 | 27.83 | 14.70 | 16.46 | 12.17 |
| NuminaMath-7B-CoT | 43.73 | 51.22 | 33.04 | 40.14 | 44.51 | 33.91 | 17.20 | 18.90 | 14.78 |
| MetaMath-13B-V1.0 | 21.15 | 32.32 | 5.22 | 7.53 | 7.32 | 7.83 | 5.73 | 4.88 | 6.96 |
| MAmmoTH2-8B | 12.90 | 11.59 | 14.78 | 17.92 | 17.07 | 19.13 | 7.53 | 10.37 | 3.48 |

lems, referred to as Original in the following subsections. We do not allow any tool usage including access to a code interpreter, as we find that many problems can be trivially solved by writing a brute-force search program.

To check whether the generated answer matches the ground-truth answer, we adopt an equivalence checker following Hendrycks et al. (2021); Shao et al. (2024), which first performs string normalization and then uses sympy package to check the equivalence of two mathematical objects.

### 3.1. Benchmarking the Performance of LLMs

We consider a wide range of language models including long-CoT models, closed-sourced large models, open-sourced small models, and math-specific models. The version information of the models is deferred to Appendix A.

In Table 1, we report the overall accuracies of the LLMs on Original, MATH-P-Simple, and MATH-P-Hard, and also separately calculate the accuracies for problems that originate from the `train` split and `test` split. As expected, for all the models we evaluate, we find that the performance on MATH-P-Hard is significantly lower than the original problems, which indicates MATH-P-Hard is more difficult.

In the meantime, most models also suffer a slight performance drop on MATH-P-Simple compared to the original problems. We note that the performance drops mainly come from the `train` split. Generalization errors still exist for the state-of-the-art models even when the test examples follow the exact same reasoning patterns as the training problems.

For problems that originate from the `test` split, ideally, both the original problem and its MATH-P-Simple modification should be equally "unseen" to the model. We observe mixed results empirically from Table 1: for gemini-2.0-flash-exp, GPT-4-turbo, claude-3.5-sonnet, the performance drops are larger than 5%, while surprisingly the performance of Phi-3.5-mini-instruct increases. For most of the models we evaluated, the accuracies on MATH-P-Simple `test` split are close to the accuracies on the original `test` split. We commend that while Srivastava et al. (2024) found a *relatively* 58% to 80% performance drop between their modified benchmark and the original MATH benchmark among a different set of the models (the best model they tested was GPT-4), we did not observe such huge gaps for the models we evaluate, which is a sign of the progress in the robustness of the newly developed models against simple perturbations.

**Inference-time Scaling.** Scaling inference-time computes has been shown to be able to boost the performance of LLMs (Wang et al., 2022; Brown et al., 2024; Wu et al., 2024; Cobbe et al., 2021b; Lightman et al., 2023). We defer the study of inference-time scaling on our benchmarks to Appendix C.5.

### 3.2. Failure Mode Analysis

To study the generalization abilities of models against hard perturbations , we focus on the set of problems where the models fail on the MATH-P-Hard modification but correctly solve either the original problem or the MATH-P-Simple modification, which accounts for 20%-47% of the total

<table>
<tr><td>

**Original**

**Problem**: Ten people are sitting around a round table. Three of them are chosen at random to give a presentation. What is the probability that the three chosen people were sitting in consecutive seats?

- - - - - - - - - - - - - - - - - - - - - - - - - - -

**Solution**:
Let's count as our outcomes the ways to select 3 people without regard to order. There are $\binom{10}{3} = 120$ ways to select any 3 people. The number of successful outcomes is the number of ways to select 3 consecutive people. There are only 10 ways to do this – think of first selecting the middle person, then we take his or her two neighbors.

Therefore, the probability is $\frac{10}{120} = \boxed{\frac{1}{12}}$.

</td><td>

**MATH-P-Hard**

**Problem**: Ten people are sitting around a round table. Three of them are chosen at random to give a presentation in a specific order. What is the probability that the first and second presenter were sitting in consecutive seats and at the same time the second and third presenter were sitting in consecutive seats?

- - - - - - - - - - - - - - - - - - - - - - - - - - -

**Claude-3.5-Sonnet**:
1) First, let's understand what we're looking for:
* We need 3 people in a specific order
* The 1st and 2nd people must be sitting next to each other
* The 2nd and 3rd people must be sitting next to each other
* This means all three people must be sitting consecutively around the table
`<-Omitting several lines->`

Therefore, the probability is $\boxed{\frac{1}{12}}$.

</td></tr>
</table>

*Figure 5.* An example of memorization *coupled with* incorrect reasoning: The model incorrectly reduces the modified condition to the original condition, and then follows the original reasoning pattern. The correct answer is $\boxed{1/36}$. We manually performed 20 repeated trials and found that Claude-3.5-Sonnet has a pass rate of 50%. Among the mistakes, 30% are due to the memorization issue above.

problems. For these problems, one can use the correct solutions to the easier problems as a reference to better determine the failure modes on the hard problems. We defer the discussion on the other cases to Appendix C.1.

First, we observe general failure modes when models are exposed to harder problems, including making mistakes in basic numerical computations and algebraic operations, making unjustified claims, missing several cases, and lacking certain math knowledge. These types of errors are more prominent in weaker models.

Besides general failure modes, when we compare the wrong solution to the MATH-P-Hard modification with the solutions to the easier versions, we are able to recognize an adequate number of memorization issues. Specifically, we found that models may **ignore the modified assumptions and presume that the original assumptions still hold**; see Figure 5 for an example. In other cases, the models may **blindly apply the techniques for the original problems** without first determining whether these techniques are still suitable in the modified setting (the responses in Figure 1 are such an example generated by GPT-4o). Interestingly, the models may even **output the desired outcome of the original problem** (not provided in the context) instead of the modified problem, e.g. Figure 6. This kind of memorization behavior is difficult to capture with most existing type of perturbations in the literature (similar to our MATH-P-Simple) that does not require different solving strategies.

These issues are often coupled with other types of errors and pervasive among the models we evaluated. For large models, we estimate the percentages of errors caused by memorization to be 40% for o1-mini and 25%

for Claude-3.5-Sonnet, via manual inspections of 20 error cases. The general failure modes due to insufficient capabilities are largely reduced for stronger models, making the memorization issues more prominent. As the capabilities of language models continue to advance, we expect the memorization issues will be the next bottleneck of reasoning models, and we urge more studies on investigating the generalization abilities of reasoning models against hard perturbations.

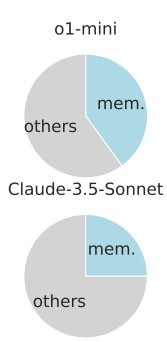

### 3.3. Is Mode Collapse a Problem?

We investigate whether the model makes errors due to mode collapse, which means the model fails to identify the difference between the perturbed problem and the original problem (seen during its training time) and the model's response *collapses* to the response to the original problem with the identical answer.

For each model, we report $n_{\text{same}}$, the number of problems where the model's final answer coincides with the ground-truth answer of the corresponding original problem. For those responses, we also compute the edit distance between the full response to the modified problem and the full response to the original problem. The full result is deferred to Table 5 in the appendix.

We see that this type of failure mode accounts for less than 10% of the total errors except for three models (gemini-2.0-flash-thinking-exp, o1-mini, and gemini-2.0-flash-exp) on

| Original | MATH-P-Hard |
|---|---|
| **Problem**: If $\lfloor n^2/4 \rfloor - \lfloor n/2 \rfloor^2 = 2$, then find all integer values of $n$. | **Problem**: If $\lfloor n^2/9 \rfloor - \lfloor n/3 \rfloor^2 = 2$, then find the smallest integer value of $n$. |

**Original**

**Problem**: If $\lfloor n^2/4 \rfloor - \lfloor n/2 \rfloor^2 = 2$, then find all integer values of $n$.

- - - - - - - - - - - - - - - - - - - - - - - - - - - - - -

**Solution:**
If $n$ is even, then we can write $n = 2m$ for some integer $m$. Substituting,

$$\lfloor (2m)^2/4 \rfloor - \lfloor (2m)/2 \rfloor^2 = m^2 - m^2 = 0.$$

Hence, $n$ must be odd; we can write $n = 2m + 1$ for some integer $m$. Substituting,

$$\lfloor (2m+1)^2/4 \rfloor - \lfloor (2m+1)/2 \rfloor^2$$

<-Omitting several lines->

$$= m.$$

Thus, we find $m = 2$ and $n = \boxed{5}$ as the unique integer solution.

**MATH-P-Hard**

**Problem**: If $\lfloor n^2/9 \rfloor - \lfloor n/3 \rfloor^2 = 2$, then find the smallest integer value of $n$.

- - - - - - - - - - - - - - - - - - - - - - - - - - -

**o1-mini:**
Let's solve the equation step-by-step:

<-Omitting several lines->

5. **Analyze for each value of $r$:**

<-Omitting several lines->

- **For $r = 1$:**

$$2 \le \frac{2k}{3} + \frac{1}{9} < 3$$

Solving gives $k = 3$ and $k = 4$. Therefore, $n = 10$ and $n = 13$.

<-Omitting several lines->

**Final Answer:**

10 and 13

*Figure 6.* An example of memorizing the desired outcome. The model outputs all integer values instead of the smallest integer value. The correct answer is $\boxed{10}$. We manually performed 20 repeated trials and found that o1-mini has a pass rate of 75%. All the 25% errors are due to this specific memorization issue above.

**MATH-P-Hard.** After manual inspection, we find that except for only 1 problem pair where gemma-2-9b-it generates the identical answer for the original problem and the modified problem, we do not see collapses of the outputs **in the superficial text form**. Therefore, we conclude that naive recitation of the *training material* is not the major reason for producing the same answers. *Instead*, the model's responses to the modified problems often collapse to the responses to the original problems **in more subtle manners**, e.g. ignoring or failing to understand the modified assumptions; see Figure 5 for an example.

### 3.4. Does In-context Learning Help or Hurt?

In this subsection, we investigate whether using the corresponding original unmodified problem and solution as the one-shot in-context learning (ICL) example will help with the modified problems in MATH-P-Simple and MATH-P-Hard. We visualize the influences of ICL for three models in Figure 7 and defer the full result to Table 6.

As expected, using the original (problem, solution) pair as a one-shot in-context demonstration boosts the performance of nearly all the models on MATH-P-Simple, which should be solvable by simply applying the original solution steps to the modified setting.

As for the MATH-P-Hard modifications, there are two factors that need to be considered: (1) **ICL effect**: the original solutions may supply the model with desired mathematical knowledge that is also helpful for solving the modified problems; (2) **misleading effect**: on the other hand, as there

are subtle differences between the original problems and the MATH-P-Hard modifications, the models may fail to recognize such differences and be misled by the demonstrated solutions. Accordingly, in Table 7 and Figure 7, and we calculate and visualize (1) $n_{\text{wrong}\rightarrow\text{correct}}$, the number of problems that initially the model fails on *without* the in-context demonstrations but answers correctly *with* the in-context demonstrations, and (2) $n_{\text{correct}\rightarrow\text{wrong}}$, the number of problems that initially the model answers correctly *without* demonstrations but fails on *with* demonstrations.

We observe that many MATH-P-Hard problems become solvable with the original problems and solutions as demonstrations. The percentages to the number of total errors without demonstrations are larger for closed-sourced large models (24%-40%) and smaller for open-sourced small models (2%-15%), due to their differences in mathematical capabilities and in-context learning capabilities. **However**, we also observe many MATH-P-Hard problems become incorrect with demonstrations, and the percentages are higher for large models (18%-40%) than small models (4%-15%). The misleading effect counteracts the effect of in-context learning, leaving only marginal improvements (less than 5%) on the MATH-P-Hard for most models.

As in-context learning can be viewed as a form of (test-time) training, we hypothesize that any naive fine-tuning technique with a limited distribution of problem settings will hurt the generalization of the language models against hard perturbations.

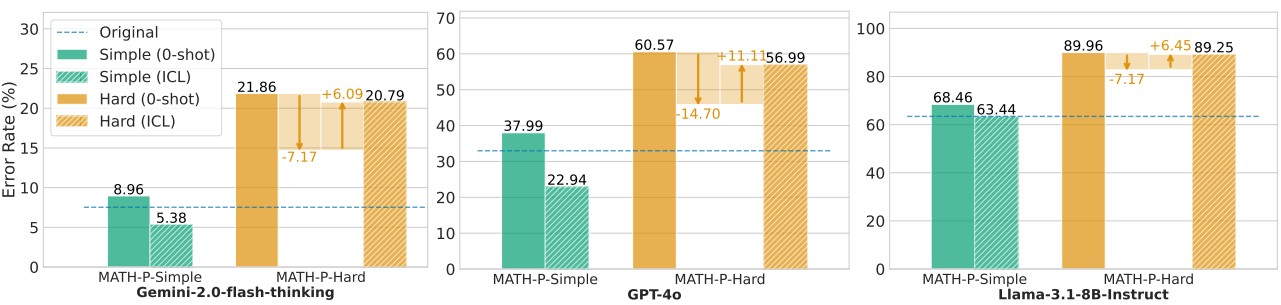

*Figure 7.* The error rates (%) of the models without and with the original problem and solution as the in-context learning (ICL) example. For MATH-P-Hard, we decompose the influences of in-context learning into **ICL effect** (the down arrow ↓), which reduces the error rates, and **misleading effect** (the up arrow ↑), which increases the error rates.

## 4. Related Work

**Perturbations to Existing Mathematical Benchmarks.** There is a considerable amount of work focusing on performing perturbations to existing mathematical benchmarks. Shi et al. (2023a) built GSM-IC from GSM8K (Cobbe et al., 2021b) by adding irrelevant context to the problem. GSM-Plus (Li et al., 2024b) creates 8 types of variations to each of the GSM8K problem and ensure that the perturbed problem is of the same difficulty. Mirzadeh et al. (2024) built GSM-Symbolic that alters the numerical values and entity names via symbolic templates of both the problems and the solution steps. Similarly, Functional MATH (Srivastava et al., 2024) is created from the MATH dataset (Hendrycks et al., 2021), and Putnam-AXIOM (Gulati et al., 2024) from the Putnam Mathematical Competition.

This line of work performed **simple perturbations** to existing mathematical benchmarks and the perturbed problems can be solved with the same solution steps and the same reasoning pattern as the original ones. In contrast, we performed **hard perturbations** to curate MATH-P-Hard, where the original reasoning pattern does not apply.

**Memorization.** Memorization is a well-studied phenomenon in machine learning (Feldman & Zhang, 2020; Zhang et al., 2021; Feldman, 2020) and has become increasingly prevalent in large language models, due to the growing of the pretraining corpora and the scaling of the model sizes. Verbatim memorization, i.e., recitation of the training material, has significant potential consequences ranging from privacy violations (Carlini et al., 2022; Brown et al., 2022; Huang et al., 2023) and copyright infringement (Shi et al., 2023b; Karamolegkou et al., 2023; Wei et al., 2024; Chen et al., 2024) to training data security risks (Carlini et al., 2021; Nasr et al., 2023). Prior work has investigated various factors influencing verbatim memorization, including sequence duplicates (Lee et al., 2021; Hernandez et al., 2022), model size (Tirumala et al., 2022), and sequence position (Biderman et al., 2023).

In contrast, we investigate the effect of memorization within the mathematical reasoning context. Our methodology falls into the category of *counterfactual tests* (Zhang et al., 2023; Wu et al., 2023; Zheng et al., 2023; Xie et al., 2024), where we construct perturbed problems different from the existing ones to test the generalization of LLMs and examine memorization effects. Through extensive case studies, we find that LLMs can exhibit subtle forms of memorization *beyond* naive verbatim memorization.

**Comparison with MATH[2] (Shah et al., 2024).** Shah et al. (2024) created MATH[2] by combining random pairs of skills extracted from MATH (Hendrycks et al., 2021) to generate harder problems that require both skills to solve. Their benchmark is mathematically harder, but there are no natural "original problems" as references. Therefore, MATH[2] is not directly suitable for investigating the memorization effects of language models. In comparison, our MATH-P-Hard are modified directly from the problems in MATH so that the modified problems require harder skills to solve. MATH-P-Hard can serve as both a harder math benchmark and a testbed to investigate memorizations of LLMs.

## 5. Conclusion

In this work, we study the generalization of large language models' math reasoning abilities against hard perturbations of the problems. We modified 279 problems from the level-5 problems of the MATH dataset (Hendrycks et al., 2021) into MATH-P-Simple (used for control experiments) and MATH-P-Hard, via simple perturbations and hard perturbations, respectively. We found performance degradations of all models on MATH-P-Hard, and many of the errors can be traced to a new form of memorization, where the model memorizes the problem-solving techniques from the training set and blindly applies them without judging whether the modified settings are still suitable. Using the original unmodified problem and solution for in-context learning can deteriorate this issue. We expect the generalization against hard perturbations to be the next major bottleneck of LLMs' reasoning abilities and urge future work in this direction.

## Acknowledgements

We acknowledge Professor Jonathan Cohen (Princeton) and Andrew Tomkins (Google) for the helpful feedback and discussion. Kaixuan Huang acknowledges the support of Google PhD Fellowship. Chi Jin acknowledges the support from the National Science Foundation NSF-OAC-2411299 and NSF-IIS-2239297. Mengdi Wang acknowledges support by NSF grants DMS-1953686, IIS-2107304, and ONR grant 1006977. The research is also supported by Princeton Language and Intelligence (PLI) Compute Cluster.

## Impact Statement

This paper introduces new mathematical benchmarks for evaluating the robustness of mathematical reasoning capabilities of large language models (LLMs). Our benchmark can help researchers identify weaknesses and guide future improvements of reasoning models, which could drive improvements in their applications such as automated theorem proving, scientific discovery, and AI-driven tutoring.

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

## A. Version Information of the Models

We consider the following models in the paper.

- long-CoT models: o1-preview, o1-mini (OpenAI, 2024), Gemini 2.0 flash thinking
- closed-source models: GPT-4o, GPT-4 Turbo (Achiam et al., 2023), Gemini 1.5 Pro, Gemini 2.0 flash (Team et al., 2024a), Claude 3.5 Sonnet, Claude 3 Opus (Anthropic, 2024);
- open-sourced general-purpose models: Llama 3.1 (Dubey et al., 2024), Gemma 2 (Team et al., 2024b), Phi-3.5 (Abdin et al., 2024);
- math-specific models: MetaMath (Yu et al., 2024), MAmmoTH2 (Yue et al., 2024), Deepseek-Math (Shao et al., 2024), Qwen2.5-Math (Yang et al., 2024), NuminaMath (Li et al., 2024a), Mathtral[1].

*Table 2.* Version information of the models

| Model | Provider | Version/Link |
|---|---|---|
| Gemini-2.0-flash-thinking-exp | Google DeepMind | 2024-12-19 |
| o1-preview | OpenAI | 2024-09-12 |
| o1-mini | OpenAI | 2024-09-12 |
| Gemini-2.0-flash-exp | Google DeepMind | 2024-12-11 |
| Gemini-1.5-pro | Google DeepMind | gemini-1.5-pro-002 |
| GPT-4o | OpenAI | 2024-08-06 |
| GPT-4-turbo | OpenAI | 2024-04-09 |
| Claude-3.5-sonnet | Anthropic | 2024-10-22 |
| Claude-3-opus | Anthropic | 2024-02-29 |
| Llama-3.1-8B-Instruct | Open-Sourced | https://huggingface.co/meta-llama/Llama-3.1-8B-Instruct |
| Gemma-2-9b-it | Open-Sourced | https://huggingface.co/google/gemma-2-9b-it |
| Phi-3.5-mini-instruct | Open-Sourced | https://huggingface.co/microsoft/Phi-3.5-mini-instruct |
| Deepseek-math-7b-rl | Open-Sourced | https://huggingface.co/deepseek-ai/deepseek-math-7b-rl |
| Qwen2.5-Math-7B-Instruct | Open-Sourced | https://huggingface.co/Qwen/Qwen2.5-Math-7B-Instruct |
| Mathstral-7b-v0.1 | Open-Sourced | https://huggingface.co/mistralai/Mathstral-7B-v0.1 |
| NuminaMath-7B-CoT | Open-Sourced | https://huggingface.co/AI-MO/NuminaMath-7B-CoT |
| MetaMath-13B-V1.0 | Open-Sourced | https://huggingface.co/meta-math/MetaMath-13B-V1.0 |
| MAmmoTH2-8B | Open-Sourced | https://huggingface.co/TIGER-Lab/MAmmoTH2-8B |

## B. Benchmark Statistics

*Table 3.* Number of problems corresponding to different subjects.

| Subject | Number (Percentage) |
|---|---|
| Prealgebra | 35 (12.54 %) |
| Algebra | 79 (28.32 %) |
| Number Theory | 36 (12.90 %) |
| Counting & Probability | 38 (13.62 %) |
| Geometry | 21 (7.53 %) |
| Intermediate Algebra | 48 (17.20 %) |
| Precalculus | 22 (7.89 %) |
| **Total** | 279 |

---

[1]Mathtral https://mistral.ai/news/mathstral/

*Table 4.* Number and percentage of the models' responses that belong to each of the four categories.

| Model | Case I | Case II | Case III | Case IV |
|---|---|---|---|---|
| Gemini-2.0-flash-thinking-exp | 212 (75.99 %) | 5 (1.79 %) | 6 (2.15 %) | 56 (20.07 %) |
| o1-preview | 194 (69.53 %) | 10 (3.58 %) | 8 (2.87 %) | 67 (24.01 %) |
| o1-mini | 218 (78.14 %) | 4 (1.43 %) | 1 (0.36 %) | 56 (20.07 %) |
| Gemini-2.0-flash-exp | 176 (63.08 %) | 11 (3.94 %) | 11 (3.94 %) | 81 (29.03 %) |
| Gemini-1.5-pro | 145 (51.97 %) | 28 (10.04 %) | 13 (4.66 %) | 93 (33.33 %) |
| GPT-4o | 94 (33.69 %) | 56 (20.07 %) | 16 (5.73 %) | 113 (40.50 %) |
| GPT-4-turbo | 81 (29.03 %) | 72 (25.81 %) | 15 (5.38 %) | 111 (39.78 %) |
| Claude-3.5-Sonnet | 88 (31.54 %) | 56 (20.07 %) | 20 (7.17 %) | 115 (41.22 %) |
| Claude-3-Opus | 49 (17.56 %) | 99 (35.48 %) | 25 (8.96 %) | 106 (37.99 %) |
| Llama-3.1-8B-Instruct | 21 (7.53 %) | 137 (49.10 %) | 7 (2.51 %) | 114 (40.86 %) |
| Gemma-2-9b-it | 22 (7.89 %) | 164 (58.78 %) | 11 (3.94 %) | 82 (29.39 %) |
| Phi-3.5-mini-instruct | 22 (7.89 %) | 161 (57.71 %) | 18 (6.45 %) | 78 (27.96 %) |
| Deepseek-math-7b-rl | 25 (8.96 %) | 138 (49.46 %) | 13 (4.66 %) | 103 (36.92 %) |
| Qwen2.5-Math-7B-Instruct | 61 (21.86 %) | 70 (25.09 %) | 15 (5.38 %) | 133 (47.67 %) |
| Mathstral-7b-v0.1 | 28 (10.04 %) | 136 (48.75 %) | 13 (4.66 %) | 102 (36.56 %) |
| NuminaMath-7B-CoT | 39 (13.98 %) | 118 (42.29 %) | 9 (3.23 %) | 113 (40.50 %) |
| MetaMath-13B-V1.0 | 6 (2.15 %) | 199 (71.33 %) | 10 (3.58 %) | 64 (22.94 %) |
| MAmmoTH2-8B | 9 (3.23 %) | 201 (72.04 %) | 12 (4.30 %) | 57 (20.43 %) |

## C. Additional Experimental Results

### C.1. Categorizing Model Responses Across Problem Variations

Recall that for each problem, we have a MATH-P-Simple modification which can be solved using the same method as the original problem, and a MATH-P-Hard modification which requires more difficult problem-solving skills. Therefore, there are 8 possible cases regarding the correctness of the model's responses to the three problems. Modulo the fluctuations of the model's correctness among the MATH-P-Simple variations, we can summarize the model's responses into the following 4 cases:

- **Case I**: at least one of the original problem and the MATH-P-Simple modification is solved *correctly*, and the MATH-P-Hard modification is also solved *correctly*.
- **Case II**: both the original problem and the MATH-P-Simple modification are solved *incorrectly*, and the MATH-P-Hard modification is also solved *incorrectly*.
- **Case III**: both the original problem and the MATH-P-Simple modification are solved *incorrectly*, but the MATH-P-Hard modification is solved *correctly*.
- **Case IV**: at least one of the original problem and the MATH-P-Simple modification is solved *correctly*, but the MATH-P-Hard modification is solved *incorrectly*.

For each of the models, we calculate the percentage of the responses in Table 4. As expected, stronger models have a higher percentage of Case I responses and a lower percentage of Case II responses. Interestingly, the percentages of Case III responses are small (less than 10%) but non-zero, where the models cannot solve the easier variants but can solve the hard variant correctly. After manual inspection, we found that this is due to the misalignment between the models' capabilities and the annotators' perception of the difficulties of math problems.

### C.2. Is Mode Collapse a Problem?

We provide Table 5 to support Section 3.3.

*Table 5.* The number of errors with answers that match the corresponding original answers. The edit distances are normalized by the length of the responses to the original problems.

| Model | MATH-P-Simple | | | | | | MATH-P-Hard | | | | | |
|---|---|---|---|---|---|---|---|---|---|---|---|---|
| | Num. Errors | | | Normalized Edit Distance | | | Num. Errors | | | Normalized Edit Distance | | |
| | $n_{same}$ | $n_{total}$ | percentage | min. | avg. | max. | $n_{same}$ | $n_{total}$ | percentage | min. | avg. | max. |
| Gemini-2.0-flash-thinking-exp | 2 | 25 | 8.00 | 0.553 | 0.611 | 0.668 | 10 | 61 | 16.39 | 0.508 | 0.679 | 0.976 |
| o1-preview | 1 | 34 | 2.94 | 0.652 | 0.652 | 0.652 | 5 | 77 | 6.49 | 0.729 | 1.07 | 1.89 |
| o1-mini | 0 | 14 | 0 | N/A | N/A | N/A | 9 | 60 | 15.00 | 0.559 | 14.7 | 126.0 |
| Gemini-2.0-flash-exp | 4 | 48 | 8.33 | 0.644 | 0.82 | 1.09 | 13 | 92 | 14.13 | 0.546 | 1.1 | 1.76 |
| Gemini-1.5-pro | 5 | 63 | 7.94 | 0.472 | 0.751 | 1.3 | 11 | 121 | 9.09 | 0.257 | 0.866 | 1.58 |
| GPT-4o | 4 | 106 | 3.77 | 0.709 | 0.773 | 0.937 | 14 | 169 | 8.28 | 0.489 | 0.777 | 1.2 |
| GPT-4-turbo | 5 | 125 | 4.00 | 0.621 | 0.74 | 0.855 | 17 | 183 | 9.29 | 0.636 | 0.932 | 1.61 |
| Claude-3.5-Sonnet | 6 | 116 | 5.17 | 0.509 | 0.729 | 0.83 | 13 | 171 | 7.60 | 0.461 | 0.741 | 1.92 |
| Claude-3-Opus | 3 | 162 | 1.85 | 0.355 | 0.485 | 0.614 | 15 | 205 | 7.32 | 0.463 | 0.841 | 1.54 |
| Llama-3.1-8B-Instruct | 13 | 191 | 6.81 | 0.595 | 0.901 | 1.99 | 18 | 251 | 7.17 | 0.618 | 0.946 | 2.7 |
| Gemma-2-9b-it | 3 | 202 | 1.49 | 0.361 | 0.506 | 0.716 | 7 | 246 | 2.85 | 0 | 0.662 | 1.08 |
| Phi-3.5-mini-instruct | 8 | 199 | 4.02 | 0.427 | 0.61 | 0.832 | 12 | 239 | 5.02 | 0.289 | 0.754 | 1.69 |
| Deepseek-math-7b-rl | 9 | 186 | 4.84 | 0.189 | 0.423 | 0.676 | 11 | 241 | 4.56 | 0.121 | 1.5 | 4.24 |
| Qwen2.5-Math-7B-Instruct | 6 | 135 | 4.44 | 0.376 | 0.591 | 0.813 | 10 | 203 | 4.93 | 0.273 | 1.01 | 4.91 |
| Mathstral-7b-v0.1 | 11 | 178 | 6.18 | 0.0989 | 0.645 | 0.964 | 13 | 238 | 5.46 | 0.105 | 0.586 | 0.984 |
| NuminaMath-7B-CoT | 12 | 167 | 7.19 | 0.241 | 0.743 | 1.62 | 14 | 231 | 6.06 | 0.204 | 1.04 | 2.22 |
| MetaMath-13B-V1.0 | 13 | 258 | 5.04 | 0.27 | 0.55 | 0.748 | 14 | 263 | 5.32 | 0.509 | 0.982 | 2.83 |
| MAmmoTH2-8B | 5 | 229 | 2.18 | 0.00214 | 0.666 | 1.25 | 9 | 258 | 3.49 | 0.708 | 0.822 | 1.04 |

## C.3. The Effect of In-Context Learning

In Table 6, we report the performance of in-context learning (ICL) with the corresponding original (unmodified) problem and solution as the in-context learning example. Furthermore, we decompose the influences on MATH-P-Hard into the **ICL effect** and the **misleading effect** in Table 7 and visualize the influences for representative models in Figure 8. Please refer to Section 3.4 for the full discussion.

*Table 6.* Performance comparisons without and with the original problem and solution as the in-context learning example.

| Model | Original (0-shot) | MATH-P-Simple | | MATH-P-Hard | |
|---|---|---|---|---|---|
| | | zero-shot | ICL w. original | zero-shot | ICL w. original |
| Gemini-2.0-flash-thinking-exp | 92.47 | 91.04 | 94.62 | 78.14 | 79.21 |
| o1-preview | 87.81 | 87.81 | 91.40 | 72.40 | 74.19 |
| o1-mini | 94.27 | 94.98 | 94.98 | 78.49 | 78.49 |
| Gemini-2.0-flash-exp | 88.17 | 82.80 | 89.96 | 67.03 | 67.38 |
| Gemini-1.5-pro | 77.78 | 77.42 | 88.17 | 56.63 | 60.57 |
| GPT-4o | 67.03 | 62.01 | 77.06 | 39.43 | 43.01 |
| GPT-4-turbo | 56.99 | 55.20 | 69.89 | 34.41 | 39.07 |
| Claude-3.5-Sonnet | 64.52 | 58.42 | 83.15 | 38.71 | 49.46 |
| Claude-3-Opus | 41.94 | 41.94 | 68.10 | 26.52 | 33.33 |
| Llama-3.1-8B-Instruct | 36.56 | 31.54 | 36.56 | 10.04 | 10.75 |
| Gemma-2-9b-it | 27.60 | 27.60 | 42.65 | 11.83 | 14.34 |
| Phi-3.5-mini-instruct | 26.16 | 28.67 | 36.92 | 14.34 | 14.34 |
| Deepseek-math-7b-rl | 37.28 | 33.33 | 45.52 | 13.62 | 15.41 |
| Qwen2.5-Math-7B-Instruct | 58.78 | 51.61 | 56.99 | 27.24 | 26.88 |
| Mathstral-7b-v0.1 | 36.56 | 36.20 | 48.39 | 14.70 | 16.49 |
| NuminaMath-7B-CoT | 43.73 | 40.14 | 47.31 | 17.20 | 17.20 |
| MetaMath-13B-V1.0 | 21.15 | 7.53 | 11.11 | 5.73 | 3.58 |
| MAmmoTH2-8B | 12.90 | 17.92 | 31.18 | 7.53 | 5.73 |

*Table 7.* Effects of in-context learning (ICL) with original example on MATH-P-Hard. The percentages of $n(\text{correct} \to \text{wrong})$ are normalized by the number of errors with ICL, while the percentages of $n(\text{wrong} \to \text{correct})$ are by the number of errors without ICL.

| Model | num. errors (zero-shot) | num. errors (ICL w. original) | $n(\text{correct} \to \text{wrong})$ | $n(\text{wrong} \to \text{correct})$ |
|---|---|---|---|---|
| Gemini-2.0-flash-thinking-exp | 61 (21.86 %) | 58 (20.79 %) | 17 (29.31 %) | 20 (32.79 %) |
| o1-preview | 77 (27.60 %) | 72 (25.81 %) | 21 (29.17 %) | 26 (33.77 %) |
| o1-mini | 60 (21.51 %) | 60 (21.51 %) | 24 (40.00 %) | 24 (40.00 %) |
| Gemini-2.0-flash-exp | 92 (32.97 %) | 91 (32.62 %) | 30 (32.97 %) | 31 (33.70 %) |
| Gemini-1.5-pro | 121 (43.37 %) | 110 (39.43 %) | 27 (24.55 %) | 38 (31.40 %) |
| GPT-4o | 169 (60.57 %) | 159 (56.99 %) | 31 (19.50 %) | 41 (24.26 %) |
| GPT-4-turbo | 183 (65.59 %) | 170 (60.93 %) | 33 (19.41 %) | 46 (25.14 %) |
| Claude-3.5-Sonnet | 171 (61.29 %) | 141 (50.54 %) | 27 (19.15 %) | 57 (33.33 %) |
| Claude-3-Opus | 205 (73.48 %) | 186 (66.67 %) | 35 (18.82 %) | 54 (26.34 %) |
| Llama-3.1-8B-Instruct | 251 (89.96 %) | 249 (89.25 %) | 18 (7.23 %) | 20 (7.97 %) |
| Gemma-2-9b-it | 246 (88.17 %) | 239 (85.66 %) | 14 (5.86 %) | 21 (8.54 %) |
| Phi-3.5-mini-instruct | 239 (85.66 %) | 239 (85.66 %) | 17 (7.11 %) | 17 (7.11 %) |
| Deepseek-math-7b-rl | 241 (86.38 %) | 236 (84.59 %) | 19 (8.05 %) | 24 (9.96 %) |
| Qwen2.5-Math-7B-Instruct | 203 (72.76 %) | 204 (73.12 %) | 32 (15.69 %) | 31 (15.27 %) |
| Mathstral-7b-v0.1 | 238 (85.30 %) | 233 (83.51 %) | 19 (8.15 %) | 24 (10.08 %) |
| NuminaMath-7B-CoT | 231 (82.80 %) | 231 (82.80 %) | 23 (9.96 %) | 23 (9.96 %) |
| MetaMath-13B-V1.0 | 263 (94.27 %) | 269 (96.42 %) | 11 (4.09 %) | 5 (1.90 %) |
| MAmmoTH2-8B | 258 (92.47 %) | 263 (94.27 %) | 12 (4.56 %) | 7 (2.71 %) |

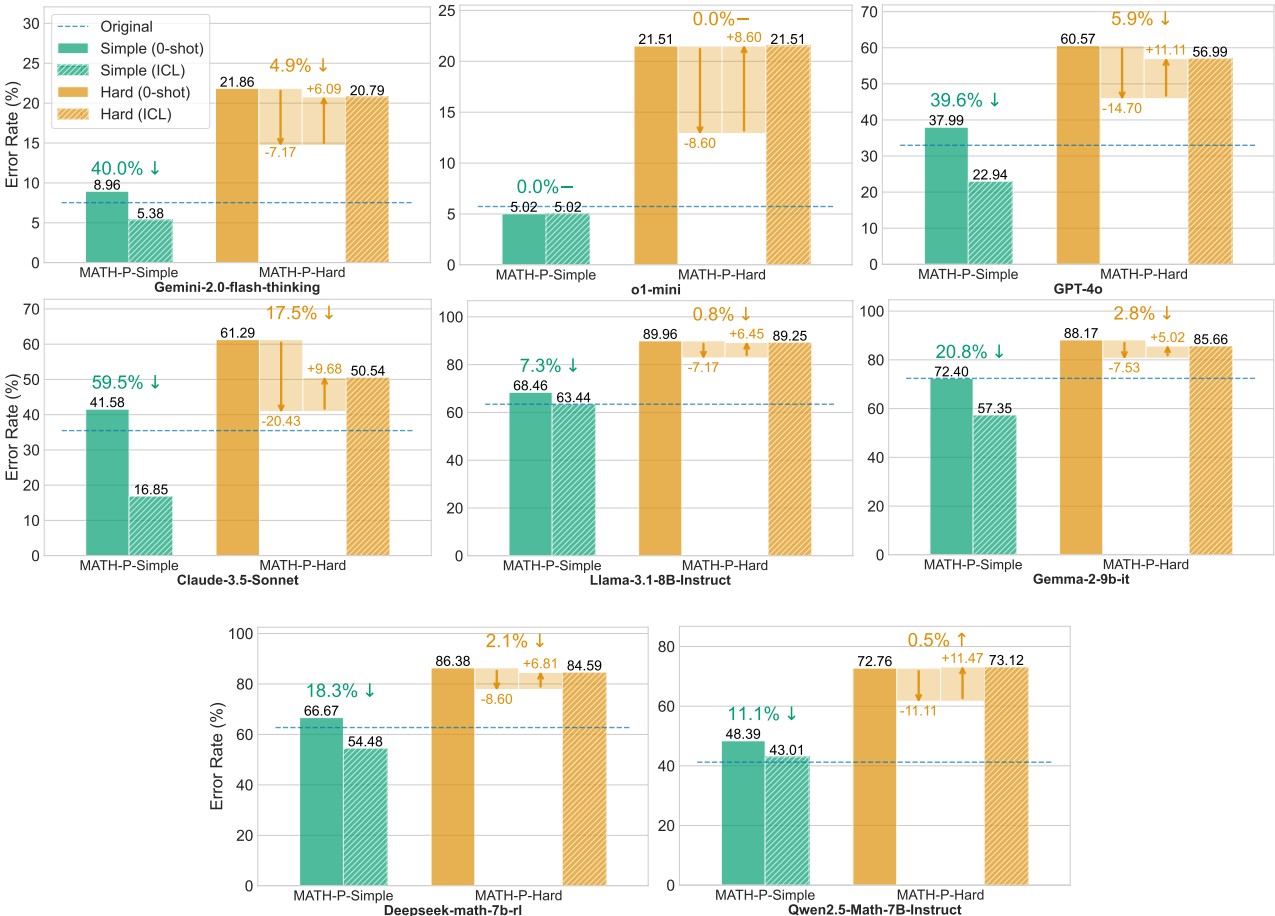

*Figure 8.* The error rates (%) of the models without and with the original problem and solution as the in-context learning (ICL) example. For MATH-P-Hard, we decompose the influences of in-context learning into **ICL effect** (the down arrow ↓), which reduces the error rates, and **misleading effect** (the up arrow ↑), which increases the error rates.

## C.4. Ablation Study: In-Context Learning with the Original Example v.s. In-Context Learning with a Random Example

In Table 8, we compare (1) the performance of one-shot in-context learning with the corresponding **original** unmodified (problem, solution) with (2) the performance of ICL with a **random** problem and solution chosen from the same category as the query problem. We find that ICL with the **original** problem and solution consistently outperforms ICL with a **random** example except for only one case.

*Table 8.* Performance comparisons without and with the original problem and solution as the in-context learning example.

| Model | MATH-P-Simple | | MATH-P-Hard | |
|---|---|---|---|---|
| | ICL w. original | ICL (random) | ICL w. original | ICL (random) |
| o1-mini | **94.98** | 92.83 | **78.49** | 75.99 |
| Gemini-1.5-pro | **88.17** | 75.99 | **60.57** | 51.97 |
| GPT-4o | **77.06** | 63.08 | **43.01** | 37.28 |
| GPT-4-turbo | **69.89** | 57.71 | **39.07** | 32.62 |
| Claude-3.5-Sonnet | **83.15** | 62.37 | **49.46** | 40.86 |
| Claude-3-Opus | **68.10** | 45.52 | **33.33** | 23.66 |
| Llama-3.1-8B-Instruct | **36.56** | 28.32 | **10.75** | 6.45 |
| Gemma-2-9b-it | **42.65** | 27.60 | **14.34** | 12.90 |
| Phi-3.5-mini-instruct | **36.92** | 20.07 | **14.34** | 10.39 |
| Deepseek-math-7b-rl | **45.52** | 34.41 | **15.41** | 13.26 |
| Qwen2.5-Math-7B-Instruct | **56.99** | 55.20 | **26.88** | 26.16 |
| Mathstral-7b-v0.1 | **48.39** | 24.37 | **16.49** | 8.96 |
| NuminaMath-7B-CoT | **47.31** | 24.73 | **17.20** | 10.04 |
| MetaMath-13B-V1.0 | **11.11** | 8.60 | 3.58 | **5.38** |
| MAmmoTH2-8B | **31.18** | 3.94 | **5.73** | 2.15 |

## C.5. Inference-time Scaling Behaviors

In this subsection, we investigate the inference-time scaling behaviors of LLMs on our benchmarks. We compute the pass@k metric following Chen et al. (2021). Specifically, for each problem, we generate $N$ solutions independently, and compute the pass@k metric via the following formula for each $1 \leq k \leq N$:

$$\text{pass@k} = \mathbb{E}_{\text{problem}} \left[ 1 - \frac{\binom{N-c}{k}}{\binom{N}{k}} \right], \text{ where } c \text{ is the number of correct answers of the } n \text{ runs.}$$

We also compute the performance of self-consistency (Wang et al., 2022), a.k.a., majority voting, where for each $k$, we randomly sample $k$ responses from the $N$ runs and get the majority-voted answer. We report the average and standard deviation among 5 random draws. We only evaluate three models: o1-mini, Llama-3.1-8B-Instruct, and Qwen2.5-Math-7B-Instruct. For Llama-3.1-8B-Instruct, and Qwen2.5-Math-7B-Instruct, we choose $N = 64$, while for o1-mini we set $N = 8$. The results are plotted in Figure 9.

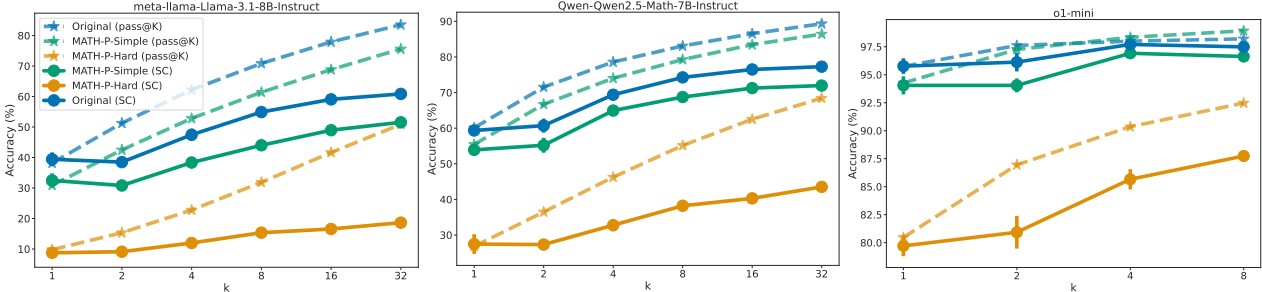

*Figure 9.* The effect of scaling up inference-time compute. We report pass@k and self-consistency (SC) accuracies for different numbers of solutions $k$.

