# OpenReview forum: "MATH-Perturb: Benchmarking LLMs' Math Reasoning Abilities against Hard Perturbations"
_ICML.cc/2025/Conference — ICML 2025 poster_

### Official Review · Reviewer_tTX2 · 2025-03-14

**Overall Recommendation:** 3

**Summary:**

This paper constructs a new dataset by applying simple an hard perturbations to the hard problems in the original MATH dataset. Experimental results show drop in performance for almost all models.

## Update after rebuttal

I remain positive about the paper after reading author rebuttal.

**Claims And Evidence:**

The claims are generally supported by evidence and convincing, though some interpretations of experimental results seem anecdotal.

**Essential References Not Discussed:**

--

**Experimental Designs Or Analyses:**

- Both train and test splits seem relatively small. Did you observe significant variance in the performance of the same model when running it multiple times?
- I am curious whether the issue in Figure 5 is repeatable or it just a one-off because of the specifics of the problem statement. Could you provide more details about your manual inspection of 20 error cases?
- Generally a lot of analysis seems anecdotal. Would it be possible to provide statistical evidence for the phenomena described?
- "We do not allow any tool usage including access to a code interpreter, as we find that many problems can be trivially solved by writing a brute-force search program." - it might still be good to evaluate with code interpreter, and then remove those problems that are trivially solvable with code as there reasoning is likely less necessary.

**Methods And Evaluation Criteria:**

Evaluation criteria make sense.

**Other Comments Or Suggestions:**

Generally it would be good to have more details on how were the experiments on failure modes performed, and have more convincing argument that it's not just anecdotal evidence based on few examples.

**Other Strengths And Weaknesses:**

I think the paper has a nice contribution as most of the existing work around this problem (generating perturbations of a dataset) falls short of creating interesting/hard perturbations.

Somehow I am not fully sure to what extent would I even consider these Hard perturbations as actual perturbations of the original dataset. Sure at the syntax level the modification is very small. But for example, in Figure 3, hard perturbation leads to a completely different solution so I would not consider these problems similar. So it's not that unexpected to me that LLMs perform worse there.

**Questions For Authors:**

Please see above.

**Relation To Broader Scientific Literature:**

The contributions are well positioned as they directly improve over prior work that considered only simple symbolic perturbations (e.g. GSM8K-Symbolic).

**Theoretical Claims:**

Yes.

---

> ### Author Rebuttal · Authors · 2025-04-01
>
> Thank you for your positive feedback on our work! **We evaluated 12 new long-CoT models that appeared near or after the ICML submission deadline.** The results [here](https://anonymous.4open.science/r/icml2025_13579_math_perturb_rebuttal-C0F8/) show no sign of saturation on MATH-P-Hard. We would like to provide detailed responses below:
>
> ---
> > **Q1**: Both train and test splits seem relatively small. Did you observe significant variance in the performance of the same model when running it multiple times?
>
> **A1**: **We did not observe significant variances in the performance.**
>
> - Our Fig. 9 contains the error bars of the performance of multiple runs, which shows the standard deviation is less than 1% (note for Fig. 9: the Self-Consistency with $k=1$ corresponds to the standard evaluation.)
> - For our new results on 12 long-CoT models, the averaged standard deviation of the performance of 3 independent runs is 0.91%.
> - One may still be concerned by the size of our benchmark. For reference, we would like to point out that the functional variants subset of Putnam-AXIOM only contains 52 problems, GSM-Symbolic only contains 100 problems, and AIME 2024/2025 contains 30 problems.
>
> ---
> > **Q2** I am curious whether the issue in Figure 5 is repeatable or it just a one-off because of the specifics of the problem statement. Could you provide more details about your manual inspection of 20 error cases?
>
> **A2**: **The memorization issue is a frequently observed phenomenon in our experiments. We did not cherry-pick the example in Figure 5.** We understand that readers may have concerns about the proportions of memorization issues, so we have quantified them manually. We plan to open-source the benchmark so our claims can be publicly scrutinized.
>
> We attach the raw logs of our manual inspections in the [anonymous github link](https://anonymous.4open.science/r/icml2025_13579_math_perturb_rebuttal-C0F8/). As the generated solutions are long and unformatted in Markdown, we omit them altogether with the problem statements and only include the problem ID, error type, and comment.
>
>
>
> ---
> > **Q3** Generally a lot of analysis seems anecdotal. Would it be possible to provide statistical evidence for the phenomena described?
>
> **A3**: **We have already taken extra caution before drawing any conclusion in the experiment section. To support each claim, we have provided quantitative numbers of different metrics as well as qualitative studies that require extensive human labor.** We hope our response to your Q2 can mitigate your concern. Besides this, could you specifically point out the claim that you think lacking statistical evidence? We are happy to address any further concerns.
>
> ---
> > **Q4**: it might still be good to evaluate with code interpreter, and then remove those problems that are trivially solvable with code as there reasoning is likely less necessary.
>
> **A4**: **We don’t think reasoning is less necessary for problems that can be trivially solvable with code.** Many number theory problems and counting problems can be solved via brute-force code solutions. However, for example, a counting problem may require the mathematical knowledge of **inclusion-exclusion principle** to solve, and a number theory problem may require knowledge of **finite group theory** to solve. The answers to these problems can often be produced by trivial brute-force codes, but they do require mathematical knowledge and reasoning ability.
>
> ---
> > **Q5** Somehow I am not fully sure to what extent would I even consider these Hard perturbations as actual perturbations of the original dataset. .... So it's not that unexpected to me that LLMs perform worse there.
>
>
> **A5**: **Unexpectedness of experimental results shouldn't undermine the contribution of the actual experiments.**
> Our results may be well-expected from your high-level conceptual argument. Nevertheless, if our experiments suggested that “LLMs are strong enough to distinguish between different perturbation types and solve these problems perfectly”, this finding could also be well-expected from an opposite but compelling argument, e.g., LLM developers have intensively built training data to cover all the perturbation cases. **Therefore, curating the benchmark out and empirically verifying the hypothesis should be a valid and important contribution.**
>
> **we believe hard perturbation is an important setting, especially when reasoning models are deployed for end users or agentic uses.** It is common for end users or agent systems to make slight changes to the inputs that fundamentally alter the questions. If the model fails to identify the changes and applies the memorized solutions, it may have bad consequences. We hope our benchmark can inspire future work in this direction.
>
> ---
> **We sincerely hope that our responses can address your concerns, and we would greatly appreciate it if you would consider raising your score of our work to a clear accept given the responses.**

---

### Official Review · Reviewer_qtvW · 2025-03-14

**Overall Recommendation:** 3

**Summary:**

This paper proposes a new benchmark by modifying 279 MATH hard problems and evaluates the popular model on these questions. They also provide various analyses of the performance on these questions.

**Claims And Evidence:**

Yes.

**Essential References Not Discussed:**

No.

**Experimental Designs Or Analyses:**

Yes.

**Methods And Evaluation Criteria:**

Yes.

**Other Comments Or Suggestions:**

No.

**Other Strengths And Weaknesses:**

Strengths:
1. The new math evaluation benchmark is valuable and needed by the community.
2. Creating and verifying problems manually is valuable and more reliable.
3. The motivation for modifying widely used MATH hard problems is well-founded.
4. The modification method of MATH-P-Hard is convincing.
5. The analysis of various phenomena is helpful.

Weakness:
1. The performance drop of SOTA models like Gemini and O1 is acceptable, indicating that the math-solving ability of SOTA LLMs is truly strong. This raises concerns that the new dataset may not be sufficiently difficult and could become outdated quickly, given the rapid advancements in math-focused LLMs.

**Questions For Authors:**

1. How do you view the concept of synthesized math problems? You may consider adding a discussion on this topic in the paper.
2. I find the boundary between ‘Perturbations’ and a new question to be somewhat vague. Your example, ‘From a line to a hyperbola,’ seems more like a completely new question rather than a perturbation. In my view, ‘Perturbations’ should focus more on adding disturbances to the problem, such as introducing irrelevant or misleading information. Defining perturbations in mathematics appears to be challenging.

**Relation To Broader Scientific Literature:**

Provide the new evaluation benchmark.

**Theoretical Claims:**

Not applicable.

---

> ### Author Rebuttal · Authors · 2025-04-01
>
> Thank you for your positive feedback on our work! **We evaluated 12 new long-CoT models that appeared near or after the ICML submission deadline.** The results [here](https://anonymous.4open.science/r/icml2025_13579_math_perturb_rebuttal-C0F8/) show no sign of saturation on MATH-P-Hard. We would like to provide detailed responses below:
>
> ---
> >**Q1** The performance drop of SOTA models like Gemini and O1 is acceptable, indicating that the math-solving ability of SOTA LLMs is truly strong. This raises concerns that the new dataset may not be sufficiently difficult and could become outdated quickly, given the rapid advancements in math-focused LLMs.
>
> **A1**:
> Thank you for raising the concern! We would like to discuss with you our thoughts and emphasize our contributions as well:
>
> - (1) **There are still around 20% of the problems (55 problems) that the SOTA long CoT models fail to solve.** So, one way to address the issue is to artificially **split the benchmark into two subsets**, for example, an “easy” set and a “difficult” set. The “difficult” subset can be used to evaluate SOTA long-CoT models while the “easy” subset can be used to evaluate small or short-CoT models. In that case, the SOTA performance on the “difficult” subset will be low, which leaves room for improvements and also saves evaluation cost.
>
> -  (2). One may still be concerned by the size of the “difficult” subset. For reference, the functional variants subset of Putnam-AXIOM [1] only contains 52 problems, and GSM-Symbolic  [2] only contains 100 problems. AIME 2024/2025 contains 30 problems. **So we believe ~55 problems are an adequate number to claim sufficient contribution. These problems can serve as the seed to curate more problems.**
>
> Reference:
> - [1] Putnam-AXIOM: A Functional and Static Benchmark for Measuring Higher Level Mathematical Reasoning
> - [2] GSM-Symbolic: Understanding the Limitations of Mathematical Reasoning in Large Language Models
>
> ---
>
> > **Q2** How do you view the concept of synthesized math problems? You may consider adding a discussion on this topic in the paper.
>
>
> **A2** Thank you for the suggestion! **In our next revision, we will add a short discussion of this future direction in the conclusion section.** We believe using synthesized math problems in training is a promising approach for improving the robustness against hard perturbations. For example, one can synthesize a training dataset with paired examples of (original problem, its hard perturbation) via hybrid methods that involve both state-of-the-art LLMs and expert-level human annotators.
>
> ---
> > **Q3** I find the boundary between ‘Perturbations’ and a new question to be somewhat vague. Your example, ‘From a line to a hyperbola,’ seems more like a completely new question rather than a perturbation. In my view, ‘Perturbations’ should focus more on adding disturbances to the problem, such as introducing irrelevant or misleading information. Defining perturbations in mathematics appears to be challenging.
>
> **A3**: We agree with you that giving a precise definition of “perturbation” may be challenging. In our paper, we use **“simple perturbation”** to refer to the cases where the reasoning patterns of the modified problem remain the same, which should be closer to the “perturbation” in your view.
>
> In contrast, for **hard perturbation**, we agree that the modified problem is essentially a new problem in the sense that the two problems have different reasoning patterns. We still call the modified problem a perturbation of the original one because they look similar superficially. **We designed MATH-P-Hard in this way to deliberately elicit memorization behaviors of the models.**
>
> Setting aside the debate of definitions, **we believe that hard perturbation is a valid and important setting, especially when reasoning models are deployed for end users or agentic uses**. It is common for end users or agent systems to make slight changes to the inputs that fundamentally alter the questions. If the model fails to identify the changes and applies the memorized solutions, it may have bad consequences. We hope our benchmark can inspire future work in this direction.
>
> ---
> **We sincerely hope that our responses can address your concerns, and we would greatly appreciate it if you would consider raising your score of our work to a *clear accept* given the responses.**

---

### Official Review · Reviewer_aCm6 · 2025-03-15

**Overall Recommendation:** 3

**Summary:**

This paper investigates the robustness of mathematical reasoning models when faced with out-of-distribution problem modifications. The authors introduce MATH-P-Simple and MATH-P-Hard, two benchmark datasets that test models under simple and hard perturbations, respectively. Their evaluation reveals significant performance drops on MATH-P-Hard, highlighting that models tend to blindly apply memorized problem-solving skills without assessing their applicability to modified contexts.

**Claims And Evidence:**

The MATH-P-Hard introduces hard perturbations that alter the reasoning path, thereby increasing problem-solving difficulty. The authors evaluate instruction-tuned MLLMs and demonstrate that these models memorize problem-solving techniques from the training set rather than genuinely adapting to problem modifications. However, prior work, such as DeepSeek-R1 [1], suggests that reinforcement learning (RL) techniques can help models reduce memorization and explore reasoning paths more effectively. A key limitation of this study is that the authors do not evaluate RL-based models, leaving an open question regarding their effectiveness in addressing memorization biases in mathematical reasoning tasks.

Perturbation benchmark has been explored by the work [2].

[1]DeepSeek-AI, DeepSeek-R1: Incentivizing Reasoning Capability in LLMs via Reinforcement Learning.
[2] Chengke Zou et. al., DYNAMATH: A DYNAMIC VISUAL BENCHMARK FOR EVALUATING MATHEMATICAL REASONING ROBUSTNESS OF VISION LANGUAGE MODELS, ICLR2025.

**Essential References Not Discussed:**

Chengke Zou et. al., DYNAMATH: A DYNAMIC VISUAL BENCHMARK FOR EVALUATING MATHEMATICAL REASONING ROBUSTNESS OF VISION LANGUAGE MODELS, ICLR2025.

**Experimental Designs Or Analyses:**

If the authors use the same technique and data engine employed for MATH-P-Hard generation to create a training dataset, and then fine-tune MLLMs on this dataset, the issue of memorizing problem-solving techniques may likely remain a bottleneck of MLLMs.

Does exposure to perturbed training data actually improve generalization, or does it reinforce memorization biases?

To truly assess whether fine-tuning on such data mitigates memorization, an ablation study comparing models trained on perturbed vs. non-perturbed datasets would be necessary.

**Methods And Evaluation Criteria:**

The authors claim to have discovered a novel form of memorization, but it is unclear what distinguishes their findings from prior observations, such as those in [1], which already highlight that models memorize solution steps without truly understanding the underlying reasoning.

The authors should explicitly define what they mean by "problem-solving techniques" in line 432.

[1] Zhang, H., et al.,  A careful examination of large language model performance on grade school arithmetic. arXiv 2024.

**Other Comments Or Suggestions:**

I have no further comments.

**Other Strengths And Weaknesses:**

The paper is well-structured and the experiments are comprehensive and easy to follow.

The study provides valuable insights into model generalization and out-of-distribution reasoning.

**Questions For Authors:**

I have no further questions.

**Relation To Broader Scientific Literature:**

This paper provides a valuable contribution by identifying new limitations in reasoning adaptability, but further research is needed to determine whether alternative training paradigms, such as RL or perturbation-based fine-tuning, can overcome these issues.

**Theoretical Claims:**

There are no theoretical claims.

---

> ### Author Rebuttal · Authors · 2025-04-01
>
> > **Q1** The authors evaluate instruction-tuned MLLMs …
>
> **A1**. We would like to first clarify that our dataset only contains textual input, and we evaluated on text-only LLMs, not Multimodal LLMs.
>
> ---
> > **Q2**: A key limitation of this study is that the authors do not evaluate RL-based models...
>
> **A2**: **We did evaluate RL-based models.**
>
> - (1) We evaluated 3 long-CoT models: Gemini 2.0 flash thinking, o1-preview, and o1-mini. These models are believed to be tuned with RL, using similar techniques as Deepseek-R1. Please note that R1 (released on 2025/01/20) is considered concurrent work.
> - (2) We additionally provided the evaluation results on the 12 new long-CoT models [here](https://anonymous.4open.science/r/icml2025_13579_math_perturb_rebuttal-C0F8), including R1.
> - (3) Besides the long-CoT models, please note that Deepseek-Math-7B-RL and Qwen2.5-Math-7B-Instruct both underwent an RL tuning stage.
> ---
> > **Q3**: Essential References Not Discussed: Perturbation benchmark has been explored by DYNAMATH [2].
>
> **A3**: **We have already discussed and properly cited the DynaMath in our submission. The contributions of the two benchmarks do not conflict with each other:**
>  - DynaMath proposes 7 different perturbation types but most of them fall into the category of simple perturbations. In contrast, our paper studies hard perturbations.
> - DynaMath focuses on multimodal mathematical reasoning settings and evaluates MLLMs, while our paper focuses on text-only settings.
>
> ---
> > **Q4**: it is unclear what distinguishes their findings from prior observations, such as those in [1], which already highlight that models memorize solution steps without truly understanding the underlying reasoning.
>
> **A4**: **Our contributions are orthogonal to [1], and our findings are different from GSM1K [1].** Specifically:
>
> - (1) First of all, **the GSM1K benchmark [1] is already saturated**, with models achieving over 95% accuracy (see the online leaderboard in https://scale.com/leaderboard/math). **The leaderboard was officially deprecated in January 2025** by Scale AI, and the benchmark does not contain newly released RL-tuned models, such as Deepseep-R1.
>
> - (2) The mechanisms of the memorizations are different: the authors of [1] stated their contribution as *“To measure the existing benchmark contamination on GSM8k, we created GSM1k, a held-out benchmark designed to match the difficulty and structure of GSM8k.”* In contrast, in our evaluation results, we showed that **naive memorization of the contaminated data is not a significant issue for the newly developed models**, and these models are already capable of generalizing to simply-perturbed problems. Instead, the memorization effects on MATH-P-Hard are caused by failing to recognize the essential differences between the perturbed problems and the original ones.
>
> ---
> > **Q5**: The authors should explicitly define what they mean by "problem-solving techniques" in line 432.
>
> **A5**: **We believe the term is already clear from the context.**
>
>  Cautions should be taken when giving a definition to such an abstract concept. By problem-solving techniques one can mean “the procedure of applying mathematical knowledge and mathematical operations” for solving a problem. This roughly corresponds to the steps of the chain-of-thought solution. Similar concepts are utilized in [[1]](https://arxiv.org/abs/2405.12205) and [[2]](https://arxiv.org/abs/2407.21009).
>
> ---
> > **Q6** ... further research is needed to determine whether alternative training paradigms, such as RL or perturbation-based fine-tuning, can overcome these issues. ... an ablation study comparing models trained on perturbed vs. non-perturbed datasets would be necessary.
>
> **A6**:
> We agree that thorough studies on the effects of RL and perturbation-based fine-tuning datasets are necessary. This is an important follow-up but is **outside the scope of this work**. As a benchmark paper, our goal is to curate a high-quality dataset and identify new  memorization issues as a current limitation of reasoning models, and encourage future studies.
>
> ---
> > **Q7** ... Does exposure to perturbed training data actually improve generalization, or does it reinforce memorization biases?
>
> **A7** Adopting the same technique to curate a training dataset with hard perturbation is a promising future direction. However, **to ensure high quality, our benchmark was curated by expert-level annotators,** which is *too costly* for constructing a large-scale training dataset. We encourage the community to explore hybrid methods to synthesize training datasets with both state-of-the-art LLMs and expert-level annotators. **Again, this is outside the scope of this work.**
>
> ---
>
> **Given your review, we believe there were major misunderstandings on our work. We sincerely hope that our responses can resolve the misunderstandings and address your concerns, and we would greatly appreciate it if you would like to re-evaluate our work given the responses.**

---

### Official Review · Reviewer_Bq4D · 2025-03-16

**Overall Recommendation:** 4

**Summary:**

The paper constructs MATH-Perturb to evaluate the math reasoning generalization of LLMs under simple and especially hard perturbations. The authors create MATH-P-Simple (279 problems) and MATH-P-Hard (279 problems) datasets from level-5 problems in the MATH dataset. Experiment results on 18 LLMs show significant performance drops on MATH-P-Hard, indicating they struggle with hard perturbations and are biased toward the original reasoning patterns. Failure mode analysis reveals that many of the errors can
be traced to a new form of memorization, where LLMs memorize the problem-solving techniques from the training set and blindly apply them without judging whether the modified settings are still suitable.

**Claims And Evidence:**

I think the generalization of mathematical reasoning abilities, and even general reasoning abilities, which this paper focuses on, is worthy of exploration. The authors summarize the situation where LLMs can answer the original questions correctly and also handle simple variations of the original questions (such as variable substitution), but fail to solve the hard variations, as a **new form of memorization**.

I have some reservations about this claim:
- If it is considered a new form of memorization, it might be categorized as memorization of problem-solving abilities. It has learned the core abilities for such problems and can solve various variations of them, but is at a loss when facing more difficult problems or may still apply previous habitual assumptions.
- This kind of memorization might be normal, simply because the model lacks the ability to solve more difficult problems. For students, they may master easy questions but be unable to solve difficult ones.
- If it is possible to design problems that are **equivalent in question type and difficulty to MATH-P-Hard** but very different from the original questions (e.g. with a large edit distance). And if the models perform better on these problems than on MATH-P-Hard, it may indicate that they habitually use the solution approach of the original questions when solving MATH-P-Hard.

**Essential References Not Discussed:**

No

**Experimental Designs Or Analyses:**

Yes, the experimental design is generally reasonable.

**Methods And Evaluation Criteria:**

Yes, I think this benchmark is valuable.

**Other Comments Or Suggestions:**

None

**Other Strengths And Weaknesses:**

** Strengths**:
- This article is well-organized and easy to understand.
- The ideas discussed are of great value.
- The constructed dataset is also valuable, which is conducive for the community to compare different difficulty variants of the same or similar problems.

** Weaknesses and Questions**:
- When the model fails to solve difficult problems, is it simply due to the model's insufficient capabilities, or should it be attributed to the model's excessive memorization?
- I think there may be a lack of some more in-depth analysis, based on the existing benchmarks in the current community. For example, could it provide ideas or clues on how to achieve easy-to-hard generalization?

**Questions For Authors:**

** Weaknesses and Questions**:
- When the model fails to solve difficult problems, is it simply due to the model's insufficient capabilities, or should it be attributed to the model's excessive memorization?
- I think there may be a lack of some more in-depth analysis, based on the existing benchmarks in the current community. For example, could it provide ideas or clues on how to achieve easy-to-hard generalization?

**Relation To Broader Scientific Literature:**

Perhaps it would be beneficial to discuss or analyze some benchmarks[1] that focus on perturbations at the level of mathematical problem-solving tasks.
[1] Zhou et al. Is Your Model Really A Good Math Reasoner? Evaluating Mathematical Reasoning with Checklist. In ICLR 2025.

**Theoretical Claims:**

Yes, please refer to "Claims And Evidence" for details.

---

> ### Author Rebuttal · Authors · 2025-04-01
>
> Thank you for your positive feedback on our work! **We evaluated 12 new long-CoT models that appeared near or after the ICML submission deadline.** The results [here](https://anonymous.4open.science/r/icml2025_13579_math_perturb_rebuttal-C0F8/) show no sign of saturation on MATH-P-Hard. We would like to provide detailed responses below:
>
> ---
>
> > **Q1**: … This kind of memorization might be normal, simply because the model lacks the ability to solve more difficult problems. For students, they may master easy questions but be unable to solve difficult ones.
>
> **A1**:
> **We believe hard perturbation is a valid and important setting, especially when reasoning models are deployed for end users or agentic uses**. It is common for end users or agent systems to make slight changes to the inputs that fundamentally alter the questions. If the model fails to identify the changes and applies the memorized solutions, it may have bad consequences, even though one can argue this kind of memorization is normal. We hope our benchmark can inspire future work in this direction.
>
> ---
> > **Q2**: If it is possible to design problems that are equivalent in question type and difficulty to MATH-P-Hard but very different from the original questions (e.g. with a large edit distance). And if the models perform better on these problems than on MATH-P-Hard, it may indicate that they habitually use the solution approach of the original questions when solving MATH-P-Hard.
>
> **A2**: Thank you for the insightful suggestion! **We designed MATH-P-Hard to mimic the original problem formulations to deliberately elicit memorization behaviors of the models.** Designing problems that are equivalent in question type and difficulty to MATH-P-Hard but with large edit distances will lead to a good subset for *isolating* the memorization effect. We agree with you that if a model solves this type of problem correctly but fails on MATH-P-Hard, we can claim that the model possesses the skills to solve the harder problem but habitually uses the memorized approach due to the superficial similarity of the problem formulation to the original one. This is an interesting **follow-up direction**.
>
> ---
> > **Q3**: Perhaps it would be beneficial to discuss or analyze some benchmarks[1] that focus on perturbations at the level of mathematical problem-solving tasks. [1] Zhou et al. Is Your Model Really A Good Math Reasoner? Evaluating Mathematical Reasoning with Checklist. In ICLR 2025.
>
>
> **A3**: Thank you for the pointer! **We will cite and discuss the paper in our next revision.**
>
> - The paper establishes a pipeline to generate perturbations of problems in 4*4 = 16 different variants, featuring 4 task generations (problem-solving, answerable judging, outcome judging, and process judging) and 4 reasoning robustness modifications (original problem, problem understanding, irrelevant disturbance, and scenario understanding). They focused on simpler GSM8K dataset and multimodal geometry datasets.
>
> - Our paper focuses on dissecting “perturbation” into simple perturbations and hard perturbations, and investigates the proportion of the failures that are due to memorization. We selected MATH level-5, which is the harder high-school competition level.
>
> ---
> > **Q4**: When the model fails to solve difficult problems, is it simply due to the model's insufficient capabilities, or should it be attributed to the model's excessive memorization?
>
>
> **A4**:  **We have discussed the failure modes in Section 3.2.** In short, the performance drops in MATH-P-Hard can be attributed to both insufficient capabilities to handle harder problems and memorization issues. The two failure modes often couple with each other. For stronger models, the general failure modes due to insufficient capabilities are largely reduced, making memorization issues more prominent.
>
> ---
> > **Q5** I think there may be a lack of some more in-depth analysis, based on the existing benchmarks in the current community. For example, could it provide ideas or clues on how to achieve easy-to-hard generalization?
>
>
> **A5**: We agree with you that the community currently lacks a more in-depth analysis of the existing benchmarks, which **motivates** our work.
>
> There are some works focusing on easy-to-hard generation using MATH dataset (e.g. train on level 1-3 problems and test on level 4-5 problems) [1]. However, for this setting, there aren’t paired data with similar problem statements but different solutions and difficulty levels. **We believe our benchmark can serve as a testbed for future studies on easy-to-hard generalization and scalable oversights.**
>
> - [1] Easy-to-Hard Generalization: Scalable Alignment Beyond Human Supervision. Zhiqing Sun, Longhui Yu, Yikang Shen, Weiyang Liu, Yiming Yang, Sean Welleck, Chuang Gan
> ----
>
> **We sincerely hope that our responses can address your concerns, and we would greatly appreciate it if you would consider raising your score of our work to a *clear accept* given the responses.**

---

> > ### Comment · Reviewer_Bq4D · 2025-04-04
> >
> > Yeah, I think Q2 is a valuable follow-up work for MATH-Perturb. And I would raise my score to 4 as I think its current contribution is suitable for publication.

---

### Decision · Program_Chairs · 2025-05-01

**Decision:**

Accept (poster)

**Comment:**

The authors propose a benchmark consisting of "hard" perturbations of the MATH benchmark. They argue that while models can do well on easy perturbations, the performance decrease is significant on the harder perturbations. The construction of the benchmark and the results are sound. Reviewers appreciated the clear description of the dataset and its utility for evaluating reasoning. At the same time, I suggest authors to put their benchmark in perspective with other recent perturbation-based datasets mentioned by reviewers and consider if they really want to call it a new kind of memorization, or just a facet of memorization that is highlighted by the current benchmark.